

# Unveiling the complex double-edged sword role of exosomes in nasopharyngeal carcinoma

Xueyan Huang and Yuedi Tang

Department of Otorhinolaryngology Head and Neck Surgery, West China Hospital of Sichuan University, Chengdu, China

## ABSTRACT

Nasopharyngeal carcinoma (NPC) is a malignancy arising from the epithelium of the nasopharynx. Given its late diagnosis, NPC raises serious considerations in Southeast Asia. In addition to resistance to conventional treatment that combines chemotherapy and radiation, NPC has high rates of metastasis and frequent recurrence. Exosomes are small membrane vesicles at the nanoscale that transport physiologically active compounds from their source cell and have a crucial function in signal transmission and intercellular message exchange. The exosomes detected in the tissues of NPC patients have recently emerged as a potential non-invasive liquid biopsy biomarker that plays a role in controlling the tumor pathophysiology. Here, we take a look back at what we know so far about the complex double-edged sword role of exosomes in NPC. Exosomes could serve as biomarkers and therapeutic agents, as well as the molecular mechanisms by which they promote cell growth, angiogenesis, metastasis, immunosuppression, radiation resistance, and chemotherapy resistance in NPC. Furthermore, we go over some of the difficulties and restrictions associated with exosome use. It is anticipated that this article would provide the reference for the apply of exosomes in clinical practice.

## INTRODUCTION

Nasopharyngeal carcinoma (NPC) is a malignancy arising from the epithelium of the nasopharynx. The International Agency for Research on Cancer reports that in 2018, there were approximately 129,000 new cases of nasopharyngeal carcinoma (*Bray et al., 2018*). The global distribution is highly uneven, with over 70% of cases occurring in East and Southeast Asia (*Bray et al., 2018*). Globally, the incidence of nasopharyngeal carcinoma has been gradually declining, with significant reductions in South and East Asia, North America, and the Nordic countries, ranging from 1% to 5% annually (*Tang et al., 2016*; *Wei et al., 2017*). In endemic areas like Hong Kong, the incidence rate has decreased steadily since the 1980s, with a total reduction of about 30% over 20 years (*Lee et al., 2003*). Guangzhou has seen average annual decreases of about 3% for men and 5% for women from 2000 to 2011 (*Li et al., 2014*), possibly due to lifestyle and environmental changes. Males have a higher incidence of nasopharyngeal carcinoma than females, with a ratio of

Corresponding author
Yuedi Tang, tangyd@hotmail.com

about 2.5 in China in 2015 (*Chen et al., 2016*). Notably, individuals from southern China maintain a high incidence even after migrating to non-endemic areas, although the second generation shows a reduced incidence. The farther the population migrates, the lower the incidence becomes (*Yu & Hussain, 2009*). This distinct geographical distribution has led to research on risk factors, which suggests that multiple factors, including EBV infection, host genetics, and environmental factors, contribute to the development of nasopharyngeal carcinoma (*Chua et al., 2016*; *Bei et al., 2016*; *Guo et al., 2009*; *Liu et al., 2016*, *2017*; *Chang et al., 2017*). Although the present treatment strategy with a combination of chemotherapy and radiation is effective, the problems of therapeutic resistance and metastatic recurrence persist as major obstacles in the management of NPC (*Chen et al., 2022*). As a result of its distinct position and lack of particular first symptoms, NPC is usually detected at the later stage, which further affects prognosis (*Bossi et al., 2021*). Hence, it is necessary to investigate appropriate pharmaceuticals and innovative biomarkers to improve the prevention, identification, and treatment of NPC.

Many different kinds of cells are able to fuse with the plasma membrane and produce vesicles called exosomes, which have a diameter of 40–100 nm. The composition of exosomes include a diverse array of compounds, such as lipids, nucleic acids, and proteins. Exosomes serve as messengers to facilitate cell-cell communication and effectively transport components to between cells for essential functions (*Wortzel et al., 2019*; *Matarredona & Pastor, 2019*; *Whiteside, 2018*). The process of exosome absorption is not haphazard but rather reliant on the receiving cells' interactions with the exosome surface proteins. To fully understand the role of exosomes in NPC, it is crucial to isolate and characterize these nanovesicles. Various techniques, such as ultracentrifugation, size-exclusion chromatography, and commercial kits, are employed to isolate exosomes from biological fluids (*Yang et al., 2019*). Once isolated, exosomes are characterized using a combination of methods, including nanoparticle tracking analysis (NTA) to determine size distribution, transmission electron microscopy (TEM) to visualize morphology, and Western blotting to identify specific protein markers like CD9, CD63, and CD81 (*Théry et al., 2018*). These techniques are essential for ensuring the purity and quality of exosomes, which is crucial for their subsequent analysis and utilization in both basic research and clinical applications.

In several malignancies, exosomes function as key players. Their potential as biomarkers is based on their ability to transport pathogenic cargos, alter tumor microenvironments, encourage angiogenesis, and assist metastasis in NPC, *etc*. The objective of this study is to investigate the pathogenic processes, possible therapeutic uses, and pertinent research on exosomes in NPC. The aim of the article is to offer a thorough analysis of important clinical features of exosome in NPC. Additionally, the therapeutic potential of exosomes as NPC diagnostic and prognostic biomarkers, and treatment targets are discussed. Lastly, we will go over the limits and potential future research areas regarding exosomes in NPC.

## AUDIENCE

This review is intended for researchers in the field of nasopharyngeal carcinoma.

## SURVEY METHODOLOGY

PubMed database was used for related literature search using the keyword "nasopharyngeal carcinoma" "exosomes" "pathogenesis" "diagnosis" and "treatment". All types of articles are included.

## OVERVIEW OF EXOSOMES

### Exosomes biogenesis and secretion

Based on their size and place of origin, three main types of extracellular vesicles have been recognized: exosomes, microvesicles (MVs), and apoptotic bodies. Typically, microvesicles (MVs) with a diameter ranging from 100 to 1,000 nm are produced by budding off the plasma membrane (*Raposo & Stoorvogel, 2013*; *Yáñez-Mó et al., 2015*). The plasma membrane of dying cells produces apoptotic bodies, which may be anywhere from 100 to 2,000 nm in diameter. Since phagocytic cells sometimes absorb them, they are not usually involved in cellular communication (*Battistelli & Falcieri, 2020*). The tiniest extracellular vesicles produced by endocytosis are known as exosomes, and their diameters usually fall between 40 and 160 nanometers (*Ha, Yang & Nadithe, 2016*). The process of endocytosis results in the formation of endosomes, which facilitate the engulfment and absorption of various substances by cells (*Elkin, Lakoduk & Schmid, 2016*; *Malm, Loppi & Kanninen, 2016*). The process of intracellular endocytosis begins with an early endosome that incorporates the plasma membrane and extracellular cargoes. When early endosomes are forming, the trans-Golgi network and the endoplasmic reticulum are all involved. At some point during its development, the early endosome becomes the late endosome. The process of endosomes undergoing additional invagination lead to the formation of multivesicular bodies (MVBs) and intraluminal vesicles (ILVs) (*Huotari & Helenius, 2011*). Exosomes are formed when MVBs combine with the plasma membrane to release ILVs (*van Niel, D'Angelo & Raposo, 2018*; *Janas et al., 2016*). On the other hand, lysosomes or autophagosomes may combine directly with MVBs.

### Exosomes isolation

*Kandimalla et al. (2021)* provided a comprehensive overview of the methods currently in use for separating exosomes and similar vesicles from plants and animals. Alternative techniques for isolating exosomes include ultracentrifugation, isoelectric precipitation, ultra-filtration, polymer-based precipitation, size-exclusion chromatography, and microfluidic operations (*Munagala et al., 2016*). Due to its simplicity and cost-effectiveness, ultracentrifugation (UC) has emerged as the preferred technique for separation and purification. Although it enhances purity, the procedure decreases the quantity of separated exosomes (*Li et al., 2017*). In order to address this limitation, an effort was undertaken to enhance the effectiveness of the exosomes separation technique and maintaining superior degree of purity and yield. In order to increase the production of exosomes, the rate of UC was reduced by employeding ultrafiltration, a process that segregates biomolecules based on their sizes. Incorporating an additional step in this procedure, however, increases its vulnerability to contamination and escalates production expenses (*Li et al., 2017*; *Lobb et al., 2015*). Size-exclusion liquid chromatography (SEC) is

a valuable technique for effectively segregating exosomes based on their sizes. By eliminating the need to rapidly pellet exosomes, this method produces a very pure product and is effective for purifying serum or plasma (*Lobb & Möller, 2017*).

## EXOSOMES: A DOUBLE-EDGED SWORD IN NASOPHARYNGEAL CARCINOMA

Exosomes play a complex double-edged sword role of in nasopharyngeal carcinoma, they are involved in the angiogenesis and growth of NPC, the metastasis of NPC, the immune response of NPC, the chemotherapy resistance and radiation resistance of NPC. Besides, exosomes also have clinical benefits and play an important role in the diagnosis, prognosis, and treatment of NPC (Fig. 1).

### The pathogenic role of exosomes in nasopharyngeal carcinoma

#### *Exosomes are involved in the angiogenesis and growth of nasopharyngeal carcinoma*

EBV infection is causally linked to the initiation of NPC pathogenesis. The development of EBV latent infection is regarded as an initial phase of tumorigenesis. During the latent phase of infection, several viral products are produced, such as EB virus nuclear antigen (EBNA) 1, EB virus-encoded latent membrane proteins (LMP) 1 and 2, and assorted incubation period mRNA. These products coexist with EBV-related exosomes (*Gallo et al., 2017*; *Canitano et al., 2013*). LMP1 is primarily synthesised during EBV infection and is strongly related to the activation and proliferation of NPC. *In vitro* studies have shown that it has many roles, including stimulating cell proliferation, and shielding cells from apoptosis (*Yoshizaki et al., 2013*). The activation of normal fibroblasts into cancer-related fibroblasts may be facilitated by the LMP1 packed by exosomes *via* the critical NF-κB pathway (*Wu et al., 2020*). Research findings indicate that LMP1 in NPC exosomes increases the expression of syndecan-2 (SDC2) and synaptotagmin-like-4 (SYTL4) through NF-κB signaling. This stimulates cell proliferation and tumor growth by activating ERK and AKT signal pathways, and induces the expression of vascular endothelial growth factor (VEGF) receptors (*Meckes et al., 2010*; *Liao et al., 2020*). The BART1 miRNAs are believed to suppress the expression of LMP1, which might promote the development of NPC malignancy (*Yoshizaki et al., 2013*). Upon integration into exosomes, LMP2 may be subsequently released into the cells (*Teow et al., 2017*). Expressed in sera and exosomes, the levels of miR-24-3p, miR-891a, miR-106a-5p, *etc.* in NPC patients vary considerably from those in healthy controls. These microRNAs affect the formation and specialization of NPC cells by suppressing the MARK1 signaling pathway. In the tumor microenvironment, exosomes play a crucial role in facilitating cell-to-cell contact. These exosomes not only promote tumor progression but also contribute to the aggravation of NPC (*Sun et al., 2018*).

Tumor angiogenesis is a vital mechanism by which tumors form a new blood vessel formation. A higher density of microvessels has been linked to an advanced stage of tumor and a negative prognosis in NPC. Numerous exosomal processes stimulate the development of blood vessels in NPC. Firstly, exosomes produced by NPC are a main

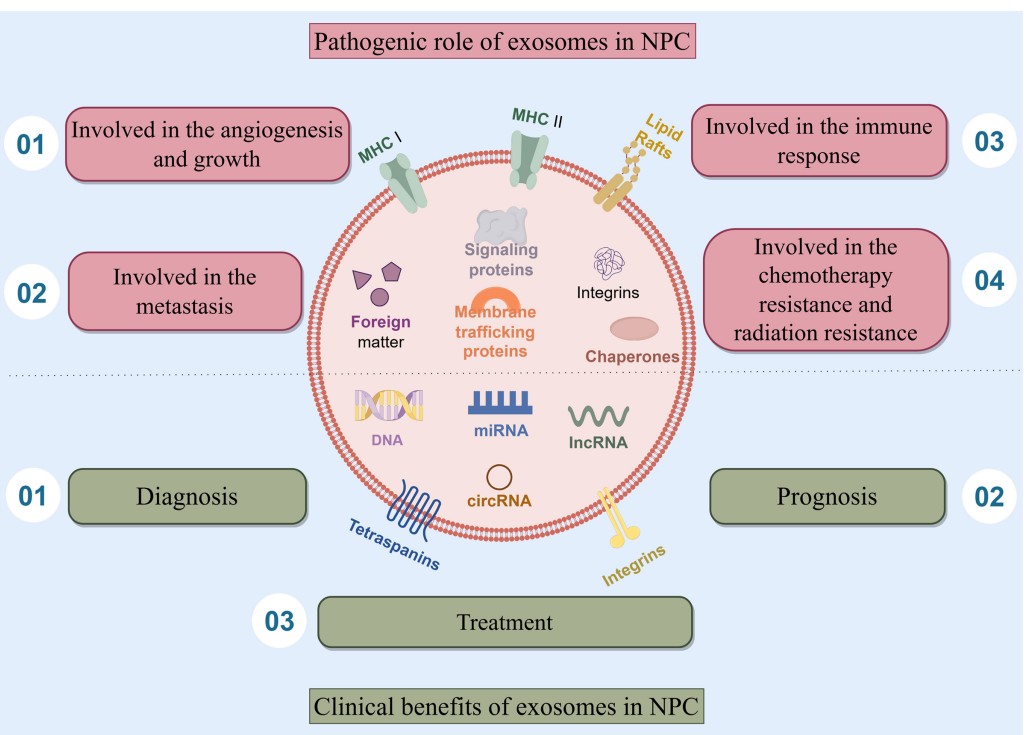

**Figure 1 The double-edged sword role of exosomes in NPC.** This article presents the main subjects addressed, which include the four pathogenic role of exosomes in NPC. Exosomes are involved in the angiogenesis and growth of NPC, the metastasis of NPC, the immune response of NPC, the chemotherapy resistance and radiation resistance of NPC. Besides, exosomes also have clinical benefits and play an important role in the diagnosis, prognosis, and treatment of NPC.

processes in stimulating the development of blood vessels. Exosomes produced from the C666-1 cell line greatly enhance the formation of tubules, movement, and invasion of HUVECs by increasing the expression of intercellular adhesion molecule-1 (ICAM-1), CD44 variant isoform 5 (CD44v5), and decreasing the expression of thrombospondin-1 (TSP-1), a protein that inhibits blood vascularization (*Chan et al., 2015*). Exosomes produced from NPC include a high concentration of HAX-1, which is positively associated with lymph node metastasis, clinical stage, M classification, unfavorable prognosis, and enhances the growth, migration, and angiogenesis. Furthermore, miRNAs may act as oncogenes inducing genetic and epigenetic alterations. MicroRNA miR17-5p released from CNE-2 cells stimulates the growth, multiplication, and movement of cancer (*Duan et al., 2019*). Survivorship in NPC patients is favorably associated with exosomal miR-9 (*Lu et al., 2018*). Endothelial tube formation and migration are inhibited *via* exosomal miR-9 *via* controlling PDK/AKT pathway and suppressing MDK (*Lu et al., 2018*) (Fig. 2).

### Exosomes are involved in the metastasis of nasopharyngeal carcinoma

Metastasis is a distinctive characteristic of cancerous tumors, characterised by a complex series of stages including cancer cells, the tumor microenvironment (TME), stromal cells, cytokines, and immune cells. Metastasis is induced by exosomes *via* several pathways. In

Huang and Tang (2025), *PeerJ*, DOI 10.7717/peerj.18783

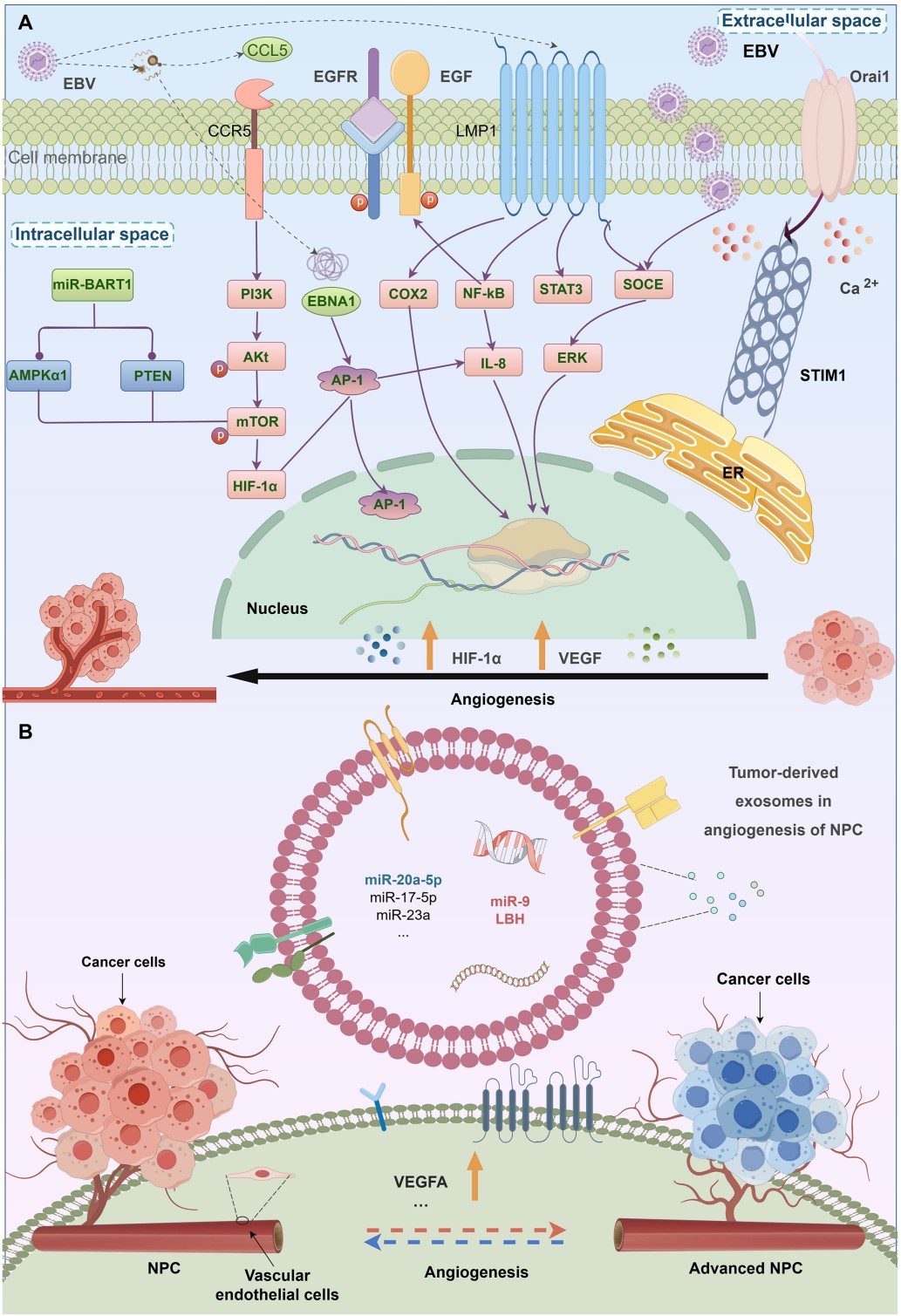

**Figure 2 The relationship between EBV infection, exosomes, and angiogenesis in NPC.** Angiogenic factors related with EBV-induced NPCs. Virus and some genes expressed by virus may stimulate angiogenesis in NPC.

the first instance, exosomes may stimulate inflammatory milieu by suppressing the immune system. Profound leukocyte infiltration and contacts between cancer and other cells are typical features (*Huang et al., 1999*). Epidermal cell carcinoma (ECV)-infected NPC cells facilitate metastasis *via* the release of cytokines and chemokines, as well as tumor exosomes, which establish communication with stromal cells and promote metastasis (*Aga et al., 2014*; *Ye et al., 2016*, *2014*; *Mrizak et al., 2015*; *Whiteside, 2015*). Furthermore, exosomes facilitate metastasis by specifically targeting receptors and ECM, hence inducing EMT, which is a process characterized by the acquisition of invasive and migratory capabilities by polarized epithelial cells, accompanied by the loss of adhesive polarity due to various stimuli. Furthermore, EMT is linked to tumor stemness, metastasis, and resistance to treatment, and takes place *via* certain intermediary phases (*Pastushenko et al., 2018*). Malignant metalloproteinases (MMPs) may be stimulated by exos that carry HIF-1α (*Shan et al., 2018*). MMP-13 is strongly expressed in the plasma of NPC patients (*You et al., 2015*). Metastasis is facilitated by exosomes that deliver MMP-13 (*You et al., 2015*). Internalisation of exosomes produced by MSCs leads to morphological modifications and modified EMT markers *via* the activation of the FGF19/FGFR4 dependent ERK signalling pathway. Internalisation of exosomes produced by MSCs leads to morphological modifications and modified EMT markers *via* activation of the FGF19/FGFR4 dependent ERK signalling pathway (*Shi et al., 2016*). A new proposal suggests that the expression of epithelial markers is terminated in the first EMT progression (*Pastushenko et al., 2018*). EVB LMP1 may decrease E-cadherin levels *via* DNA methyl transferase or transcriptal suppression, therefore facilitating EMT (*Horikawa et al., 2007*; *Tsai et al., 2002*; *Martin et al., 2005*; *Horikawa et al., 2011*). Moreover, exosomes have regulatory influence on many cellular pathways that enhance the process of metastasis. Genetic suppression of exosome release significantly reduces both multidirectional cell migration in living organisms and chemotaxis in laboratory settings (*Sung et al., 2015*) (Fig. 3).

### Exosomes are involved in the immune response of nasopharyngeal carcinoma

A significant characteristic of NPC is the invasion of a substantial quantity of non-malignant white blood cells, mostly T lymphocytes with a minor presence of B cells, macrophages, and dendritic cells. Nevertheless, the infiltration of leukocytes has been seen to vanish throughout the course of metastasis, and instead, they are substituted by quickly and extensively multiplying malignant cells that possess a clear defense mechanism against tumors (*Kapetanakis, Baloche & Busson, 2017*). Cytokines and chemical compounds produced by exosomes may cause local buildup of regulatory T cells and enhance the aggressiveness of NPC (*Gourzones, Barjon & Busson, 2012*). Galactin-9, an immunomodulatory protein found in EBV-infected NPC exosomes initiating apoptosis in fully developed CD4+ cells (*Klibi et al., 2009*). The expression of LMP1 may stimulate the production of galectin-9, leading to the liberation of exosomes that contain both LMP1 and galectin-9. The combination of them can powerfully inhibit T cells growth, unlike the synergistic effect of galectin-9 without LMP1 (*Keryer-Bibens et al., 2006*). Additionally, it can induce the suppression of Tregs by suppressing FGF11 (*Ye et al., 2016*). Interleukin-6

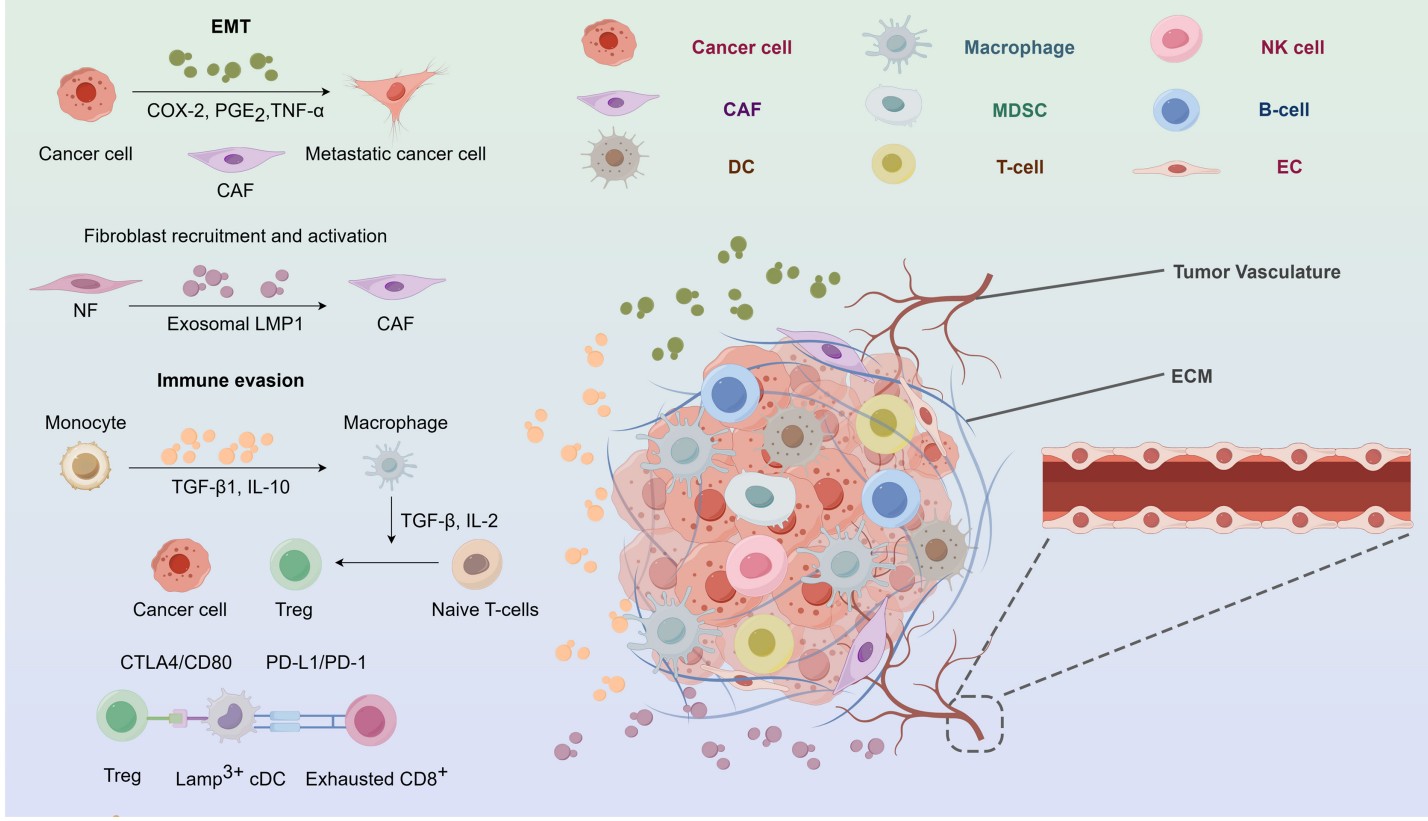

**Figure 3 The nasopharyngeal carcinoma tumor microenvironment.** CAF, cancer-associated fibroblast; COX-2, cyclooxygenase-2; CTLA4, cytotoxic tlymphocyte associated protein 4; DC, dendritic cell; EC, endothelial cell; ECM, extracellular matrix; EMT, epithelialmesenchymal transition; Exo-LMP, exosome packaged latent membrane protein; IL, interleukin; MDSCs, myeloid-derived suppressor cells; NF, normal fibroblast; NK, natural killer; PD-1, programmed cell death protein 1; PD-L1, programmed death-ligand 1; PGE2, prostaglandin E2; TGF, tumor growth factor; TNF, tumor necrosis factor; Treg, regulatory T-cell.

(IL-6) is a proliferation initiator for many types of cancer. Tregs, as the key molecules responsible for immune suppression, may serve as facilitators of tumor immune evasion. Reports indicate that the exosomal chemokine CCL20 may attract Tregs, stimulate T cells convert into inhibitory Tregs, strengthen the suppression of Tregs (*Mrizak et al., 2015*). Furthermore, infection with gamma herpesvirus may alter the protein composition of exosomes in B cells (*Meckes et al., 2013*). Collectively, exosomal chemicals have the ability to impact T cell function, sustain ongoing EBV infection, and trigger immunosuppression (Fig. 3).

### Exosomes are involved in the chemotherapy resistance and radiation resistance of nasopharyngeal carcinoma

NPC recurrence rates vary from 15% to 58% after first chemoradiotherapy. The responsiveness to radiation or chemotherapy in relapsed tumors reduced. Hence, the chemoradioresistance and recurrence of NPC emerged as the most complex challenges faced by doctors. Exosomes may enhance treatment resistance by disseminating proteins, miRNAs, and lncRNAs that promote resistance phenotype into susceptible cancer cells (*Kalluri, 2016*). The exosome contents produced by fibroblast or mesenchymal stem cells

associated with NPC can be transported from the donor to recipient cells. These exosomes regulate several pathways related to drug resistance by inhibition of immune surveillance, and elimination of chemotherapeutic drugs and radiation (*Chen et al., 2014*; *Zheng, 2017*). MSC exosomes may stimulate resistance to 5-FU by counteracting 5-FU-induced cell death and enhancing the production of MDR, MRP, and LRP-related proteins (*Ji et al., 2015*). Furthermore, the EBNA1 found in exosomes associated to EBV may disrupt miR-200a and miR-200b, therefore facilitating resistance to drugs (*Wang et al., 2014*). LMP1 in exosomes generated from EBV infection may activate the PI3K/AKT pathway to promote the stemness and resistance to chemotherapy of NPC (*Yang et al., 2013*, *2016*). Exosomes regulate EMT *via* interacting with and depositing exosomal cargo (such as DNAs, mRNAs, miRNAs) into the cells they target (*Henderson & Azorsa, 2012*; *Steinbichler et al., 2017*). For instance, NPC-Exo facilitates Erythropoiesis of cells by transmitting mitotic cytokines (such as TGF-β), resulting in resistance to chemotherapy (*Dongre & Weinberg, 2019*; *Andarawewa et al., 2012*). EBV-BART-miRNAs transported by exosomes drive EMT by specifically targeting the key tumor suppressor, phosphatase, and PTEN (*Cai et al., 2015*). Furthermore, NPC-associated exosomes may also stimulate the proliferation of fibroblasts, leading to a desmoplastic reaction that hampers the effective administration of anti-cancer medications and thereby contributes to drug resistance (*Quail & Joyce, 2013*). Chemoresistance is linked to dysregulation of miRNAs in cancer-associated fibroblasts (CAFs), which could be transferred from cancer cells to the microenvironment by exosomal transport (*Santos et al., 2016*). Exosomes produced by fibroblasts may activate cancer stem cells, which aid in the development of resistance to chemotherapy (*Hu et al., 2017*). By inhibiting AKT pathway, LMP-1 in exosomes increase CD44 level and result in radiation resistance (*Yang et al., 2014*; *Mei et al., 2014*). The function of exosomes in the development of resistance to cancer drugs is multifaceted and requires more investigation.

## Clinical benefits of exosomes for nasopharyngeal carcinoma

### The role of exosomes in the diagnosis and prognosis of nasopharyngeal carcinoma

While there has been a relative decrease in the occurrence of NPC, the early identification of this condition remains difficult because of its unusual symptoms and concealed site. A considerable number of individuals with NPC are already at an advanced stage when they are diagnosed, which exacerbates the overall prognosis. Timely detection and intervention are crucial for the prognosis of NPC. Owing to the significant impact of EBV on NPC, many EBV assays targeting EBV-DNA have been produced. However, their sensitivity and specificity are insufficient to meet the requirements of clinical use (*AbuSalah et al., 2020*; *Tan et al., 2020*). Therefore, investigating new biomarkers and techniques for early non-proliferative cell carcinoma detection becomes urgent. The use of exosomes in diagnosis has garnered much interest at present (*Jalalian et al., 2019*; *Vaidyanathan et al., 2018*). The distinctive biogenesis of exosomes provides them with the capacity to freely move in physiological fluids as carriers of numerous molecules (*Whiteside, 2018*). Recent studies highlight the potential of biomolecules found in NPC exosomes as innovative diagnostic tumor biomarkers, leveraging sophisticated technological platforms that

specifically target nanoparticle detection. Intracellular long non-coding RNA (lncRNA) expression in infected cells, exosomes, and tumors may be differentially induced by EBV infection in NPC, indicating the possible therapeutic use of lncRNA as a biomarker (*Zhang et al., 2020*). Exosomal circularity the correlation between MYC and radiation tolerance has been established, and ROC analysis indicates that MYC has the capability to differentiate radiation tolerant NPC patients from those that are sensitive (*Luo et al., 2020*). Exosomal molecules may be evaluated in conjunction with the current means of detecting EBV. *Kadriyan et al. (2021)* investigated the types of p53 carried by nasopharyngeal cancer (NPC)-derived exosomes (NPC-Exo) in NPC patients. It was found that NPC-Exo could potentially carry not only wild-type but also mutant-type p53. By isolating exosomes from serum samples of eight NPC patients, followed by RT-PCR and sequencing analysis, the study detected p53 mutation in one patient sample. This discovery provides important insights into the role of p53 in NPC and has significant implications for the future utilization of exosomes in NPC diagnosis and treatment, although further research is required to explore the clinical impact of mutant-type p53 as an exosome cargo. In the diagnosis of NPC, it has been shown that cyclophilin A (CYPA) in exosomes is much greater than that in entire sera (*Liu et al., 2019*). Furthermore, exosomes have been utilized as biomarkers and as a contrast agent for H2O2-responsive catalytic photoacoustic imaging (PAI) in NPC (*Ding et al., 2019*). Significantly, the molecules present on the surface of exosomes have been used to mark the exosome system in order to identify its release for the purpose of tumor surveillance.

Comprising tumor-derived exosomes, the liquid biopsy platform enables molecular and genetic profiling of parent tumor cells, as well as providing information on tumor state, monitoring treatment responses, assessing prognosis, and evaluating the immunological cells' capacity to induce anticancer action. Furthermore, there is a strong correlation between the concentration of exosomes in circulation and a negative prognosis for patients (*Ye et al., 2014*). Elevated levels of peripheral exosomes in individuals with NPC patients are associated with worse lymphatic metastasis and prognosis (*Ye et al., 2014*). Furthermore, exosomal miR-9 is strongly related to the negative prognosis in patients (*Lu et al., 2018*). A significant correlation exists between exosomal HMGB3 and NPC metastasis, suggesting a potential new foundation for treatment in metastatic patients (*Zhang et al., 2021*). The reduced lifespan of mice with xenografts were associated with exosomes rich in EGFRs produced by highly metastatic NPC cells. These exosomes may increase the ability to metastasize and decrease levels of intracellular ROS *via* the PI3K/AKT pathway (*Li et al., 2020*). Cancer cells generate very immunosuppressive exosomes that have a propensity to reprogramme immune cells, therefore influencing the prognosis. Exosomes produced from TW03 elevate the activity of the proinflammatory cytokines (*Ye et al., 2014*). Exosomal miR-24-3p levels regulate NPC *via* suppressing FGF11, and thus are strongly associated with reduced disease-free lifespan (*Ye et al., 2016*). Exosomes carrying EBV products may function as diagnostic indicators and treatments for NPC. Significantly elevated levels of circulating exosomal circMYC are seen in NPC. It has a positive correlation with cell survival and a negative correlation with susceptibility to radiation (*Luo et al., 2020*). Decreased plasma miR-9 levels are strongly linked to more advanced

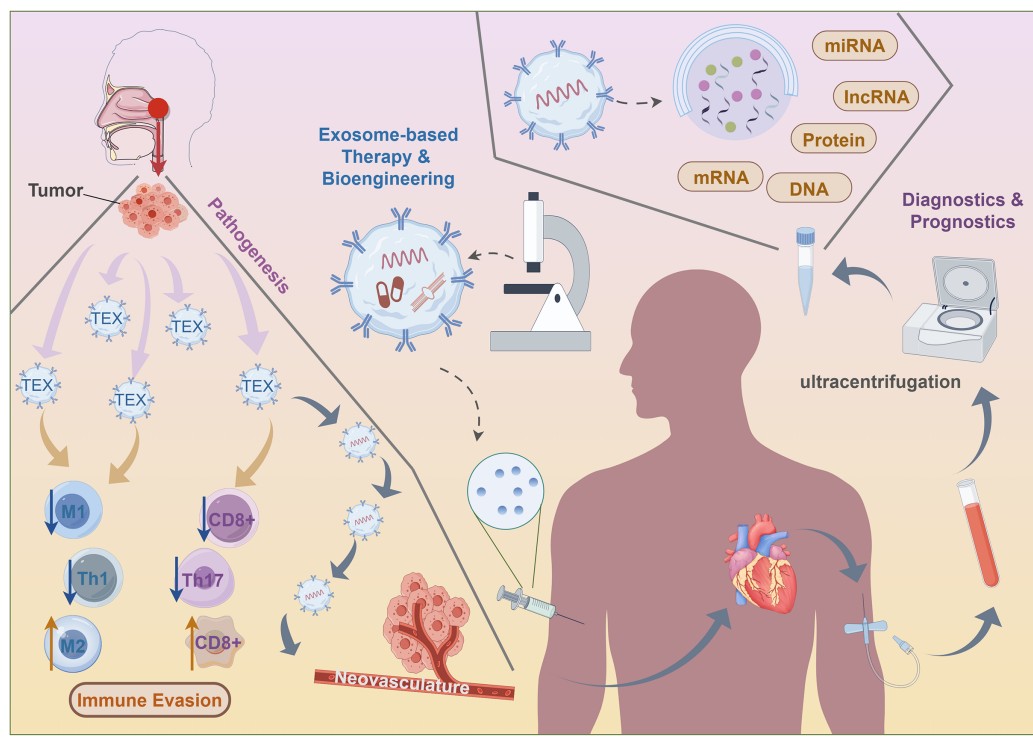

**Figure 4 Clinical application of exosomes for nasopharyngeal carcinoma.**

stages of TNM (*Lu et al., 2014*). The sensitivity and specificity of miR-9 in differentiating locoregional from metastatic NPC patients are well-established (*Choo et al., 2018*). Moreover, plasma miR-9 are markedly increased after therapy (*Lu et al., 2014*) (Fig. 4).

### The role of exosomes in the treatment of nasopharyngeal carcinoma and insights from other tumors

Exploiting naturally existing exosomes produced by the immune cells of the host also offers a promising opportunity for immunomodulation. Previous research has extensively investigated the possible apply of exosomes produced from M1 macrophages (M1-EX) as supplementary agents in therapy. The concurrent use M1-EX and PD-L1 inhibitors shown a substantial decrease in tumor growth compared to the exclusive administration of either drug (*Choo et al., 2018*). Moreover, the research revealed that the combination of M1-EX with vaccination resulted in a more robust response of particular CTLs to the antigen (*Zhu et al., 2018*). This response also shown superiority over other vaccination potentiators (*Zhu et al., 2018*). Furthermore, M1-EX could augment the effectiveness of other therapies such as cancer vaccines, as well as chemotherapy. Another area of focus in this discipline is on exosomes produced by NK cells (NEX). The first demonstration by *Zhu et al. (2019)* showed that NEX inherently includes chemicals such as TNF-α, which allow them to trigger apoptosis. The researchers then designed nanovesicles that imitate NEX by directing NK cells with increasingly reduced hole diameters, resulting in entities referred to as NK-EM (*Di Pace et al., 2020*; *Kang et al., 2021*). Significantly, NK-EM demonstrated

superior cytotoxicity compared to NEX in trials against glioblastoma, and hepatic carcinoma (*Wang et al., 2022*). Subsequent investigations revealed that introducing IL-15 to the NK cells prior to NEX extraction might further improve the ability of NEX to target tumors and stimulate cell death, hence broadening its potential applications in immunotherapy (*Zhu et al., 2019*). These method has the potential to improve the responsiveness of NPC to adoptive cytotoxic T-cell based treatment as well as other types of immunotherapy (*Wang et al., 2022*).

The following paragraphs will explore the current progress in exosome-based therapy in different types of cancer and the possibility of including them into NPC therapy. Based on the similarity of the tumor-infiltrating microenvironment (TIME) of NPC to the cancer types described before (*Chen et al., 2020*; *Gong et al., 2021*), with the exception of some small subgroups such double negative B cells (*Chung et al., 2023*), we propose the possibility of developing comparable treatment approaches for NPC. Here, we provide a concise overview of the existing exosomal approaches realized for the treatment of NPC, in order to demonstrate the feasibility of this approach.

An ongoing approach to enhance anti-tumor immunity using exosomes is their incorporation into cancer vaccines. A cancer vaccination is a therapeutic intervention that administers tumor-associated antigens (TAAs) to patients in order to instruct the immune system to selectively identify and eliminate cancerous cells (*Lin et al., 2022*). Exosomes produced by dendritic cells (DCs) have the potential to transport the MHC-antigen complex to additional inactive DCs in peripheral lymph nodes. Therefore, they could enhance the capacity of stimulation of T lymphocytes (*Taieb, Chaput & Zitvogel, 2005*). Accordingly, a clinical experiment was conducted to investigate the efficacy of DEX in augmenting an immune response against cancer. Advanced patients had good tolerance to the vaccination, and half of them showed significantly enhanced NK lytic activity (*Morse et al., 2005*).

Due to the intrinsic existence of TAAs in TEX, they may potentially serve as cancer vaccines, in addition to DEX (*Gu et al., 2015*; *Zhang & Yu, 2019*). Enhanced production of IL-12 in colon cancer TEX resulted in greater effectiveness in inhibiting tumor development and activating the immune system, when combined with a DC-based vaccine (*Rossowska et al., 2019*). Moreover, subsequent studies have shown that the augmentation of DCs by the use of TEX would ultimately lead to the revitalization of T cells (*Asadirad et al., 2019*; *Zuo et al., 2020*). In order to improve the capacity of dendritic cells to acquire and deliver tumor-associated antigens (TAAs), TEX may be bioengineered and used as cancer vaccines. Transmission electron transfer (TEX) transports a range of protumour and immunosuppressive compounds. Erroneously targeted cells may lead to severe outcomes, particularly in the case of NPC-TEX, which also contains EBV oncogenic proteins. Presently, there is a lack of specialized bioengineering methods for the elimination of certain TEX content (*Zhang et al., 2023*). Notably, it has been shown that DEX has a greater capacity to stimulate CTLs in comparison to TEX as cancer vaccines (*Rossowska et al., 2019*; *Zuo et al., 2020*; *Morishita et al., 2017*). Hence, DEX may be a more auspicious avenue than TEX in the future development of NPC therapies.

The stability and cancer-cell targeting effectiveness of exosomes make them suitable for use as a medication or chemical delivery mechanism to directly reverse immunosuppression. A dual delivery strategy for pancreatic cancer was created using exosomes produced from bone marrow mesenchymal stem cells (BM-MSCs) (*Zhou et al., 2021*). Electroporation was used to load galectin-9 siRNA into the exosomes, which were then modified with oxaliplatin prodrug (*Zhou et al., 2021*). While using of bioengineering for the development of immunotherapeutic drugs in NPC is still in its early stages, the growing body of data from related research is drawing attention. STING is a transmembrane protein found in the endoplasmic reticulum that may be expressed as TEX. The STING-rich targeted exosomes have the ability to attract TBK1 in order to stimulate IRF3 in the cells (*Gao et al., 2022*). This stimulates the synthesis of many cytokines, among which type I interferons (IFNs) are prominent. Experimental evidence has shown that activated STING, delivered *via* TEX, may stimulate the synthesis of IFNβ in macrophages. This, in turn, can attract CD3+ and CD8+ T lymphocytes to the TME (*Gao et al., 2022*). Induction of miR-6750 by manual means resulted in an elevation of its expression in TEX. TEX with high levels of miR-6750 have shown anti-tumor activity by stimulating M1 polarization in macrophages and suppressing angiogenesis (*Zhang et al., 2023*). From a therapeutic perspective, bioengineered exosomes saturated with miR-6750 have the potential to be used as an adjunct to immunotherapy due to their capacity to counteract immunosuppressive effects of tumor-associated macrophages. Nevertheless, more endeavors are necessary to determine the efficacy of these medicines in NPC, as well as to discover alternative candidate compounds that might be used in exosome delivery methods for NPC therapy. These innovative approach highlights the potential of exosome-based immunotherapy and, presumably, offers meaningful information that future studies in NPC might use (Fig. 4).

# CONCLUSIONS, LIMITATIONS, AND FUTURE PERSPECTIVES

## Potential of exosomes as biomarkers in NPC

In NPC, exosomes exert influence on the TME, contribute to resistance to chemotherapy and radiation, trigger immunological suppression, stimulate abnormal angiogenesis, and facilitate dissemination. Consequently, they have the potential to serve as valuable biomarkers. The possibility exists for their development into liquid nano-biopsy platforms based on exosomes, which may be used for cancer detection, prognosis, and appraisal of therapy responses.

## Challenges in exosome extraction for clinical applications

However, the shift from the bench to the bedside encounters some obstacles. An inherent constraint of exosomes in clinical applications is the challenges associated with their extraction. In order to fully exploit the capabilities of liquid biopsies based on exosomes, superior purity exosome isolation from patient physiological fluids, including blood, must be achieved, and there must be efficient and cost-effective methods for this.

## Current methods and limitations of exosome isolation

The goal is to produce DEX/TEX for use in cancer vaccines or to grow drug-delivering pure exosomes from mesenchymal stem cells. On the other hand, how to isolate and purify tumor exosomes from whole exosomes is the primary challenge in using exosomes for surveilling tumor progression. It will be necessary to develop cost-effective techniques for isolating exosomes from the source. Recommendations from the International Society for Extracellular Vesicles indicate that currently, there is not a highly efficient method for separating exosomes that works everywhere (*Théry et al., 2018*). Both SEC and differential UC are categorized as having "intermediate recovery, intermediate specificity," and ISEV reports that the majority of published studies have used both methods. Due to their limitations, intermediate techniques are often associated with low yields and the risk of sample contamination from other free particles.

## Emerging techniques for exosome separation

While SEC is a time-consuming method (*Takov, Yellon & Davidson, 2019*), recent studies have shown that exosomes in UC are susceptible to physical manipulation and shear stress during the multistep centrifugation procedure (*Mol et al., 2017*). The expansion of exosome therapeutic application in NPC and other cancer types requires improvements in separation methods and the integration of analytical tools into devices like microfluidic chips. Acoustic-based separation, a technique that differentiates particles of various sizes by applying different acoustic stresses, has shown potential as an innovative approach (*Wu et al., 2017*). Though less complex techniques like ultrafiltration also have promise, it is important to enhance their productivity (*Shu et al., 2020*).

## Safety considerations and potential hazards of exosome treatment

Another constraint is the safety considerations associated with exosome-based treatment. Thus, the potential hazards linked to exosome treatment remain mostly unidentified. A potential issue that may arise is contamination. The majority of exosome delivery methods now under development use MSCs as their base. *Kalluri (2016)* created a bioreactor to continuously generate high-quality exosomes (*Mendt et al., 2018*), but if the exosomes being used are not well isolated and sterilised, they may retain genetic material derived from the MSCs of origin, which may be hazardous.

## Immunosuppressive effects and purity concerns

Exosomes derived from MSCs have the potential to modulate the activities of several immune cell types by reducing the secretion of cytokines that promote inflammation (*Zhang et al., 2014*; *Chen et al., 2016*). In addition, MSC-derived exosomes may impair Th1 or Th17 cell proliferation and promoting CD4+ T cell growth into Tregs and Th2 (*Del Fattore et al., 2015*; *Duffy et al., 2011*). Within the framework of cancer therapy, these immunosuppressive effects might have adverse consequences. Moreover, the surfaces of exosomes are often altered to enable preferential fusion with target cells. Although the targeted distribution of exosomes has shown efficacy (*Alvarez-Erviti et al., 2011*), their specificity may be compromised by unforeseen events such as surface peptide breakdown,

leading to adverse consequences (*Hung & Leonard, 2015*). The existence of microRNAs and cargos within TEX that might cause cancer or promote the growth of pre-existing cancer cells makes the purity and specificity problem much the more concerning when this material is used in treatments like cancer vaccinations (*Ghamloush et al., 2019*; *Yin et al., 2021*).

## Unexplored facets of tumor immunity and non-tumor-derived exosomes

Insufficient knowledge of exosome-cellular interactions is another significant constraint. The role of exosomes in a yet unexplored facet of tumor immunity is one area within NPC that deserves particular attention. Research on various types of cancer has expanded its focus to include NK cells, MDSCs, *etc*. In contrast to early studies on NPC that mostly focused on T cells and macrophages and how TEX affects them (*Olejarz et al., 2020*; *Hao et al., 2022*; *Li et al., 2021*; *Salimu et al., 2017*). Non-tumor-derived exosomes not produced by tumors also have a substantial impact on the advancement of tumors and their ability to evade the immune system, a factor that has been neglected in the study of NPC. In different malignancies, exosomes produced from tumor may contribute to tumor immune evasion (*Marar, Starich & Wirtz, 2021*; *Xie et al., 2022*), non-tumor-derived exosomes may potentially impact tumor growth and metastasis (*Dai et al., 2020*; *Sun et al., 2020*). Future investigations of NPC should aim to broaden these avenues and refrain from an undue emphasis just on TEX.

## Future directions and prospects for exosome-based therapies

While the detection and use of exosomes encounter difficulties with sensitivity and specificity, the ongoing development of valuable tools and procedures for exosome detection and identification may provide potential solutions to these issues. Furthermore, such novel technologies are very simple and cost-effective. Given the advancements in technology, it is reasonable to anticipate the extensive use of exosomes in the diagnosis and treatment of NPC. In order to establish a standard protocol for assessing diagnosis and disease prognosis, it is necessary to gather preclinical data and conduct preclinical investigations. Those comprehensive investigations may aid in the development and refinement of a novel diagnostic and therapeutic instrument for the management of NPC. Given the growing focus, future advancements on exosome-based therapeutic approaches are anticipated to enhance the clinical treatment of NPC. Future efforts must solve concerns related to efficiency, purity, complexity, costs, and scalability in order to enable the widespread use of exosome-based liquid biopsy.

### Funding

The authors received no funding for this work.

### Competing Interests

The authors declare that they have no competing interests.

## Author Contributions

- Xueyan Huang conceived and designed the experiments, performed the experiments, analyzed the data, prepared figures and/or tables, authored or reviewed drafts of the article, and approved the final draft.
- Yuedi Tang conceived and designed the experiments, performed the experiments, analyzed the data, prepared figures and/or tables, authored or reviewed drafts of the article, and approved the final draft.

## Data Availability

This is a literature review.

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
