# Peer review of "Unveiling the complex double-edged sword role of exosomes in nasopharyngeal carcinoma"

_PeerJ, doi:10.7717/peerj.18783_

## Round 0.1 · original submission · Major Revisions

All of the issues raised by the reviewers should be addressed.

Reviewer 1 ·

Basic reporting

No comment.

Experimental design

In “3.2.2 The role of exosomes in the treatment of nasopharyngeal carcinoma”, most of the evidence or contents discussed in this part are not based on NPC, but come from other cancers. The authors may modify the subtitle of this part. I also suggest adding another subtitle like “Insights from other tumors……” to make distinguishing easier.

Validity of the findings

In “4. Conclusions, limitations, and future perspectives”, it is just too long and hard to read. The authors had better add some subtitles for this part. How to isolate and purify tumor exosomes from whole exosomes is the primary challenge in using exosomes for surveilling tumor progression, and the authors should focus on this point.

Additional comments

1) The definition of nasopharyngeal carcinoma (NPC) is not precise. The authors claim that “NPC is a malignant epithelial tumor located in the posterolateral nasopharynx”. NPC is a malignancy arising from the epithelium of the nasopharynx. However, not all the NPC “located” the posterolateral nasopharynx.
2) Please pay attention to the case-sensitive switching for “nasopharyngeal carcinoma” throughout the entire manuscript. In addition, in Line 55, a space is lacking.
3) Figure 3. Please re-size fonts appropriately and use only one uniform font, such as Arial.

·

Basic reporting

Overall the writing was excellent, however, in lines 219-221, there is an ambiguous sentence.

Experimental design

The study design was explained clearly.

Validity of the findings

The finding was good, however, it needs to be adjusted.
For instance, the role of P53 in exosome cargo was not explained in the prognosis section (pages 288-247). The author may refer to the reference below:
Kadriyan et al, NPC-exosome carry wild and mutant type p53 among nasopharyngeal cancer patients. Indones Biomed J; 2021:3(4);403-408.

Additional comments

1. I proposed to delete Figure 1 because the biogenesis and components of exosomes have been published in many journals, except there is a new finding in this manuscript.
2. A new Figure is needed to express the other pathways of exosomes that affect the NPC, not only angiogenesis, to express its pathogenic role as shown in Figure 3, but also its clinical efficacy, for instance in the treatment effect. Therefore, there will be a balanced view of its double-sword effect.

·

Basic reporting

No comment.

Experimental design

No comment.

Validity of the findings

no comment

Additional comments

he paper provides a comprehensive review of the role of exosomes in nasopharyngeal carcinoma (NPC). The authors have effectively discussed the mechanisms by which exosomes contribute to tumor progression, metastasis, and drug resistance. Additionally, the potential of exosomes as biomarkers and therapeutic agents has been well-explored.
1. Here's a suggested paragraph to introduce the topic of exosome isolation and characterization within the Introduction:
"To fully understand the role of exosomes in NPC, it is crucial to isolate and characterize these nanovesicles. Various techniques, such as ultracentrifugation, size-exclusion chromatography, and commercial kits, are employed to isolate exosomes from biological fluids. Once isolated, exosomes are characterized using a combination of methods, including nanoparticle tracking analysis (NTA) to determine size distribution, transmission electron microscopy (TEM) to visualize morphology, and Western blotting to identify specific protein markers like CD9, CD63, and CD81. These techniques are essential for ensuring the purity and quality of exosomes, which is crucial for their subsequent analysis and utilization in both basic research and clinical applications."
This paragraph can be placed after the initial introduction to exosomes, where you discuss their general characteristics and functions.
By incorporating this information into the Introduction, you can provide readers with a solid foundation for understanding the subsequent discussions on exosome biology and their role in NPC. It will also highlight the importance of rigorous isolation and characterization techniques in exosome research.
2. While NPC is prevalent in Southeast Asia, it is particularly common in southern China, especially in Guangdong and Guangxi provinces. Please provide a more accurate and detailed overview of the geographical distribution and risk factors associated with NPC.

---

## Round 0.2 · Minor Revisions

Please address two minor issues raised by the 2nd reviewer.

Reviewer 1 ·

Basic reporting

no comment

Experimental design

no comment

Validity of the findings

no comment

Additional comments

no comment

·

Basic reporting

The revised version was better than the first version

Experimental design

The type of publication included for analysis in this study should be stated in the survey methodology section. For instance, only RCT articles are included, or all types of articles are included.

Validity of the findings

The findings were good

Additional comments

1. I found the word "CYPA", but I didn't find its meaning
2. The legend of Figure 4, is better if changed to "Clinical application of exosomes for nasopharyngeal carcinoma". The reason is that the figure not only describes the benefits but also the negative impact of exosomes.

·

Basic reporting

No comment.

Experimental design

No comment.

Validity of the findings

No comment.

Additional comments

The author has adequately addressed all raised concerns, and I recommend the paper for publication.

---

## Round 0.3 · accepted · Accept

All of the issues raised by the reviewers are now addressed.